# Joint Optimization Method of Spare Parts Stocks and Level of Repair Analysis Considering the Multiple Failure Modes

**Ruiqi Wang [1], Guangyu Chen [1,*], Jie Wu [1], Wei Zhou [2] and Zheng Huang [1]**

1   School of Management and Economics, University of Electronic Science and Technology of China (UESTC), Chengdu 610054, China; ricky4018038@126.com (R.W.); wujie.hx@chinaccs.cn (J.W.); 202021150119@std.uestc.edu.cn (Z.H.)
2   The Laser Fusion Research Centre, China Academy of Engineering Physics (CAEP), Mianyang 621054, China; zkm_zky@yahoo.com.cn
*   Correspondence: chenguangyu@uestc.edu.cn

**Abstract:** For the repair level and spare parts stocking problems, generally METRIC type methods and Level of repair analysis (LORA) are used separately. Since LORA does not consider the availability of capital goods, solving LORA and spare parts stocking problems sequentially may lead to suboptimal solutions. On these considerations, this study presents a joint optimization method to minimize the service logistics cost under the constraints of system availability. Maintenance capability factor and maintenance decisions are introduced into the joint optimization model to express the influence of multiple failure modes on repair level and spare parts stocking. Thus, we establish the bridge relationship between LORA and METRIC models. The joint optimization model is solved by an improved iterative algorithm, and a typical fleet system is taken as an example to verify the correctness and effectiveness of the model and the algorithm. Compared with the optimization of spare parts inventory and maintenance level independently, the joint optimization method could effectively reduce the service logistic system cost.

**Keywords:** fleet; service logistics; LORA; VARI-METRIC; multiple failure modes

## 1. Introduction

A system is considered dynamic when its characteristics and logical or capacity configuration change over time [1]. An emblematic case of these systems is fleet systems, in which several configurations may satisfy the same goal [2]. In aircraft groups, for example, this fleet system assigns combat, training, and other missions in different geographical regions [3,4]. A small improvement in inventory costs or maintenance capability can produce meaningful savings of fleet systems. These motivations are capable of defining and quantifying the performance of a service logistics system, without losing sight of its complex characteristics.

Level of repair analysis (LORA) and spare parts stocks allocation are current topics in the field of service logistics research. LORA is used to analyze the maintenance decision of key components in the system when they fail. It could formulate the maintenance level of repair, move to repair, or discard, as well as identify maintenance resources [5]. Spare parts are critical resources of service logistics system which play an important role in operating missions. The fleet system's availability would be degraded if key components fail or spare parts are not available [6]. LORA and spare parts stocks are typically optimized individually to obtain the best cost within their individual limitations. Many researchers have neglected the relationship between spare parts stocks and maintenance, which leads to the non-optimal cost of the fleet system [7]. Therefore, it is necessary to propose a joint optimization method of spare parts stocks and level of repair analysis.

Modeling LORA in isolation usually requires an explicit relationship between components and maintenance resources such as one-to-many link [8,9], one-to-one link [10,11],

or many-to-one link [12,13]. However, there are few hypotheses about component failure modes in the previous studies. Failure modes of fleet's components are complicated, because of high speed, heavy load, or strong impulse conditions [14,15]. Maintenance resources and cost implications of different failure modes are different, such as when a fleet system is in the state of short circuit, the corresponding corrective repair time distribution may not be identical [16,17]. Therefore, multiple failure modes should be considered simultaneously for analyzing the repair level of the fleet's components.

Modeling the joint optimization of spare parts stocks and LORA, the structure of operating system and service logistics system needs to be defined separately. Alfredsson builds a nonlinear integer programming model based on the single-echelon repair network and two-indenture system structure [18]. The system unavailable time, spare parts waiting time, and fault repair time are taken into account in this model. Basten applies the enumeration method to the maintenance decision flow model to solve the feasible solution of a single-indenture system [19], and then develops the system to a two-indenture structure and suggests an iterative algorithm for handling the joint problem of spare parts stocks and LORA [20]. For the multi-echelon multi-indenture system, some researchers started seeking the approximate solution of the joint optimization model. Triki investigates the estimated domain spectrum of decision variables by simulating parameter value changes [21]. Based on the fuzzy method, Xue proposes a multivariable fuzzy parameter convex optimization algorithm to solve the EBO (Expected Back Order) problem [22]. By comparing with the convex optimization algorithm and sequential methods, Guo demonstrates the iterative algorithm superior [23]. In reference [19], the authors acknowledge two drawbacks in their work. The first is related to considering a symmetrical repair network (i.e., making the same decisions at all locations of a given echelon for each component and resource). The second is related to the simplifying assumption of infinite repair capacities. These are also the main shortcomings of current research.

Maintenance capability is an important parameter of multi-echelon multi-indenture system optimization models; only one of them is usually considered in the modeling. Liu et al. formulate maintenance capability uncertainty as a chance-constrained framework [24]. One important conclusion of this research is that a higher maintenance capability level ensures a higher availability of spare parts. Using the LORA decisions as an input, the spare parts stocking problem is solved to decide which components to put on stock at which location(s) in the repair network in which quantity, and many examples can be found in reference [25], such that a target availability of the capital goods is achieved against minimum holding costs [20].

Maintenance decision-making is also important in the multi-echelon multi-indenture system optimization models. Maintenance decisions determine inventory changes at different maintenance locations. A well-known method to solve the inventory dynamic change problem is (VARI-)METRIC [26,27], which is a greedy heuristic that is known to find solutions that are close to optimal, such as Feng et al. build a single-echelon multi-indenture joint optimization model by considering the ratio of maintenance decisions (repair, move to repair, discard) [28]. Although current research shows that joint optimization has more advantages in cost saving than single optimization, the lack of comprehensive analysis of maintenance capability and maintenance decisions would result in a solution that is not optimal.

Given these antecedents, a new two-echelon two-indenture nonlinear joint optimization model for a fleet system is built in this study. Under the availability constraint, we formulate maintenance capability factor and maintenance decisions as the influences of different failure modes. The inventory problem, repair level, and maintenance resource allocation of spare parts are solved by an improved iterative algorithm. Therefore, an economic and effective service logistics system optimal method can be established to ensure the long-term and stable economic operation of the fleet system.

## 2. System Description

This section describes the characteristics of the fleet system and service logistics system. According to the current research, we propose several hypotheses to provide the theoretical basis for the joint optimization modeling.

### 2.1. Two-Indenture Fleet System Description

A fleet system is thought to be made up of dozens or hundreds of homogeneous subsystems; each subsystem has the same structure and can be considered an independent system. In order to improve maintenance efficiency, the modular design is generally adopted. The fleet system structure would be divided into multiple indentures.

The 2-level structure is shown in Figure 1, including Shop Replaceable Unit (SRU), which is the subset of Line Replaceable Unit (LRU). Moreover, at the location of maintenance for a system, SRU can fail and be repaired. Basten has done a detailed analysis of the definition of LRU and SRU [19].

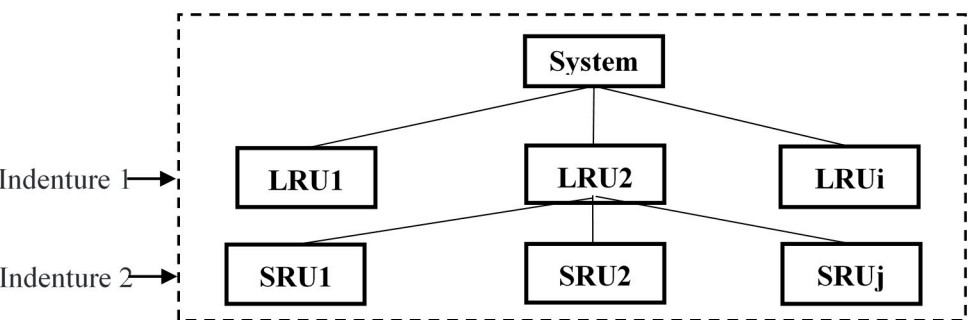

**Figure 1.** Two-indenture system structure.

The assumptions of the two-indenture fleet system are as follows [20]:

(1)   In a given fleet system, the deterioration process of each homogenous component is independent and uniform, which indicates that each component randomly degenerates according to the same law over time.

(2)   The subcomponent will be replaced to repair system, if the corresponding component fails. Therefore, when a component should be maintained, each subcomponent at the same echelon level should be decided moved, discard, or repaired.

(3)   It takes little time on replacing a defective LRU, and the replacement belongs to operational availability. Therefore, compared with the supply and operational availability, we should focus on the former one (supply availability).

(4)   Discarding the LRU implies that its SRUs are discarded as well.

(5)   For each component at each location, an $(s-1, s)$ continuous review inventory control policy (one for one replenishment) is used. That is, the replenishing and ordering strategy for one missing component is adopted to keep the stock at the level of $s$. Because each component has multiple failure modes, and each mode has a different failure mechanism and failure rate, the system has different degrees of influence and consequences [14]. Two assumptions of multiple failure modes of components are added in this study:

(6)   The failure modes of the component appear randomly.

(7)   Minor fault mode could be repaired on base location; serious fault mode should be sent to the depot location for repair.

### 2.2. Two-Echelon Service Logistics System Description

A two-echelon service logistics system is composed of three Base locations and two Depot locations. As indicated in Figure 2, the fleet system executes duties at the Base locations.

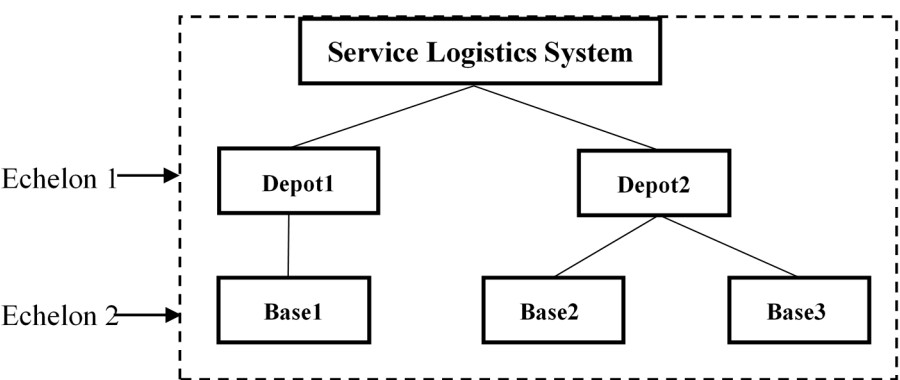

**Figure 2.** Two-echelon service logistics system.

The assumptions of the two-echelon service logistics system are as follows [20]:

(1) Each maintenance location has a set of resources, including spare parts, maintenance workers, maintenance equipment, and tools. The maintenance capability is proportional to the resources available for maintenance.
(2) The discard decision is only considered in the Depot.
(3) A failed (sub)component may not be shipped to a lower echelon level. When a failed component with echelon level $e$ should be repaired, the corresponding subcomponent must be maintained at echelon level $f \geq e$, rather than a lower level.
(4) There are no lateral transshipments between locations at the same echelon level or emergency shipments from locations at a higher echelon level; functioning spare parts are only supplied from one specific location at the next higher echelon level.

When the Base location cannot repair the failure components, they must be transferred to the Depot location for maintenance. At this point, the Depot location must decide whether to repair or discard the failed component. Due to the (*s*-1, *s*) continuous review inventory control policy, minimizing the expected number of backorders is a good approximation of maximizing the availability [26].

### 3. Two-Echelon Two-Indenture System Modelling

*3.1. Mathematical Model of LORA*

The characteristics of fleet system repair level and spare parts allocation are evaluated in this section. We introduce maintenance capability factor and maintenance decisions (repair, move, discard) into LORA and METRIC models. This method can reflect the influence of multiple failure modes on the joint optimization model.

We consider the structural dependence relationship between system components, as well as the maintenance resources in the model. LORA is used to analyze the optimal repair level of components and determine the type and quantity of maintenance resources. Therefore, the key issues of repair level analysis are as follows:

(1) Analyze the maintenance decision of failure components based on failure mode: move, repair, discard;
(2) Analyze the level of repair decision-making;
(3) Determine the maintenance resources needed to implement the maintenance decision.

According to the above two-echelon two-indenture system structure logic analysis, an LRU and its SRUs are taken as an example to build a LORA model. The variables and descriptions involved in this paper are shown in Table 1.

**Table 1.** NOTATION.

| Variable Name | | Implication |
|---|---|---|
| **Input Variables** | $i$ | Component number, LRU: $i = 0$; SRU: $i = 1 \cdots I$ |
| | $h$ | LRU family number, $h = 1 \cdots H$ |
| | $j$ | Location number, Depot: $j = 0$; Base: $j = 1 \cdots J$ |
| | $k$ | Subsystem number, $k = 1 \cdots N$ |
| | $\lambda_{i,j}$ | Yearly demand means value of component $i$ at location $j$ |
| | $q_i$ | Conditioned failure rate of a SRU on the parent LRU |
| | $r_{i,j}$ | Repairing probability of component $i$ at location $j$ |
| | $T_{i,j}$ | Repairing time of component $i$ at location $j$ |
| | $O_{i,j}$ | Order and ship time of component $i$ from the Base to the location $j$ |
| | $fc_i$ | Price of component $i$ |
| | $M_{i,j}$ | The number of components $i$ in the system |
| **Intermediate Variables** | $N_{i,j}$ | Quantity of component $i$ in repair at location $j$ |
| **Decision Variables** | $X_{i,j,d}$ | Maintenance policy d of component $i$ at location $j$ |
| | $Y_{r,j}$ | Allocation of Maintenance resources $r$ at location $j$ |
| | $s_{i,j}$ | Level of inventory of component $i$ at location $j$ |

(1) If the LRU fails, the possible maintenance decisions are repair, move, and discard:

$$X_{0,j,repair} + X_{0,j,move} = \lambda_{0,j}, j = 1, 2, \cdots \tag{1}$$

$$X_{0,0,repair} + X_{0,0,discard} = \sum_{j=1}^{J} X_{0,j,move} \tag{2}$$

Constraint (1) states that the number of repairs and moves at Base location is equal to LRUs' annual failure times.

Constraint (2) states that the number of repairs and discards at Depot location is equal to the number of LRUs moves.

(2) If the failure is determined to be caused by SRU, there are:

$$X_{i,j,repair} + X_{i,j,move} = \lambda_{i,j} \tag{3}$$

$$\sum_{j=1}^{J} X_{i,j,move} = \sum_{d \in D} X_{i,0,d} \tag{4}$$

$$X_{0,j,repair} = \sum_{i=1}^{I} X_{i,j,repair} \tag{5}$$

Constraint (3) states that the number of repairs and moves at Base location is equal to SRUs' annual failure times.

Constraint (4) states that the number of repairs and discards at Depot location is equal to the number of SRUs moves.

Constraint (5) states that the number of LRU repairs at Base location is equal to the sum of all SRUs repairs.

(3) Maintenance decisions need the corresponding maintenance resources. So $d = repair$ needs the following constraint:

$$X_{i,j,repair} \leq Y_{r,j} \cdot \lambda_{i,j}, i \subset \Omega_r \tag{6}$$

where $\Omega_r$ is a collection of all the components that need the resource $r$ for repair.

*3.2. Mathematical Model of Spare Parts Stock*

Spare parts are components of a system unit that can be dismantled and replaced immediately following a failure. According to the basic METRIC model, Base location

should refill supplies to satisfy the demand with new or fixed LRU, when an LRU is replaced on a subsystem. Two levels including Base and Depot are used to satisfy the demand of LRU, based on the inspection capability at Base level [29]. The aim of inspection is to verify the replacement and demand of SRU. Furthermore, the SRU demand at the Depot is always satisfied by repairing the failed SRU, while at the Base it can be satisfied only if the Base is certified to execute the repairing process, otherwise, the SRU demand is redirected to the Depot [30].

The total demand $\lambda_{i,j}$ of an SRU is the sum of two contributions:

$$\lambda_{i,0} = \sum_{j=1}^{J} \lambda_{i,j}(1 - r_{i,j}) + \lambda_{0,0}q_i, i = 1, 2, 3 \ldots \tag{7}$$

The demand of SRU deriving from a failure of LRU, inspected at the Bases, not certified to execute the maintenance operations:

$$\lambda_{i,j} = \lambda_{0,j} \cdot r_{0,j} \cdot q_i, i = 1, 2, 3 \ldots, j = 1, 2, 3 \ldots \tag{8}$$

The demand of SRU deriving from a failure of LRU inspected at the Depot:

$$\lambda_{0,0} = \sum_{j=1}^{J} \lambda_{0,j}(1 - r_{0,j}) \tag{9}$$

Sherbrooke points out that the mean variance ratio of the number of repairs is greater than 1, so the Poisson distribution assumption is not applicable [26]. We believe that in the discrete distribution function, the negative binomial distribution satisfies the feature that the variance is greater than the mean. It can be used to describe the discrete feature of the arrival of spare parts demand.

At the Depot, the pipeline of any SRU is composed by the components in maintenance and the Expected Backorder. The expected value and the variance of the pipeline are related only to the repairing time and to the level of spare parts in stock:

$$E[N_{i,0}] = \lambda_{i,0}T_{i,0} + EBO(s_{i,0}|\lambda_{i,0}T_{i,0}), i = 1, 2, \cdots \tag{10}$$

For each SRU, any Base offers a pipeline consisting of components under maintenance, previously requested and arriving from the Depot, and the EBO:

$$E[N_{i,j}] = \lambda_{i,j}[((1 - r_{i,j})O_{i,j} + r_{i,j}T_{i,j})] + f_{i,j}EBO(s_{i,0}|\lambda_{i,0}T_{i,0}) \tag{11}$$

$$Var[N_{i,j}] = \lambda_{i,j}[((1 - r_{i,j})O_{i,j} + r_{i,j}T_{i,j})] + f_{i,j}(1 - f_{i,j})EBO(s_{i,0}|\lambda_{i,0}T_{i,0}) \\ + f_{i,j}^2 VBO(s_{i,0}|\lambda_{i,0}T_{i,0}) \tag{12}$$

where $f_{i,j}$ is the fraction of *i*-th SRU at the Depot to supply the *j*-th Base.

$$f_{i,j} = \lambda_{i,j}(1 - r_{i,j})/\lambda_{i,0}, i = 1, 2, 3 \ldots \tag{13}$$

Analogously, the LRU pipeline at the Depot can be described by:

$$E[N_{0,0}] = \lambda_{0,0}T_{0,0} + \sum_{i=1}^{I} f_{i,0}EBO(s_{i,0}|\lambda_{i,0}T_{i,0}) \tag{14}$$

$$Var[N_{0,0}] = \lambda_{0,0}T_{0,0} + \sum_{i=1}^{I} f_{i,0}EBO(s_{i,0}|\lambda_{i,0}T_{i,0}) + \sum_{i=1}^{I} f_{i,0}(1 - f_{i,0})VBO(s_{i,0}|\lambda_{i,0}T_{i,0}) \tag{15}$$

where $f_{i,0}$ is the fraction of the *i*-th SRU needed to repair the LRU.

$$f_{i,0} = \lambda_{0,0}q_i/\lambda_{i,0} \tag{16}$$

$$\sum_{j=0}^{J} f_{i,j} = 1$$

The values representing the LRU pipeline at the Bases are then:

$$E[N_{0,j}] = \lambda_{0,j}[(r_{0,j}T_{0,j} + (1 - r_{0,j})O_{0,j})] + \sum_{i=1}^{I} EBO(s_{i,j}|E[N_{i,j}], Var[N_{i,j}]) + f_{0,j}EBO(s_{0,0}|E[N_{0,0}], Var[[N_{0,0}]) \quad (17)$$

$$\begin{aligned} Var[N_{0,j}] &= \lambda_{0,j}[(r_{0,j}T_{0,j} + (1 - r_{0,j})O_{0,j})] + f_{0,j}(1 - f_{0,j})EBO(s_{0,0}|E[N_{0,0}], Var[[N_{0,0}]) \\ &\quad + f_{0,j}^2 VBO(s_{0,0}|E[N_{0,0}], Var[[N_{0,0}]) + \sum_{i=1}^{I} VBO(s_{i,j}|E[N_{i,j}], Var[N_{i,j}]) \end{aligned} \quad (18)$$

where $f_{0,j}$ is:

$$f_{0,j} = \lambda_{0,j}(1 - r_{0,j})/\lambda_{0,0}, j = 1, 2, 3, \cdots \quad (19)$$

When the number of components in repair obeys the negative binomial distribution, The EBO of LRU at Base $j$ is:

$$EBO(s_{0,j}) = EBO(s_{0,j}|E[N_{0,j}], Var[N_{0,j}]) \quad (20)$$

where $E[N_{0,j}], Var[N_{0,j}]$ is the mean and variance of a negative binomial distribution. In order to calculate the EBO of components, we need to convert the mean and variance into parameters $(a, b)$ from the negative binomial distribution $nbin(x, a, b)$.

Since it cannot be guaranteed under any conditions, the variance of the number of parts under repair of all parts in the system at all maintenance points in the service logistics system is greater than 1. Instead of using a general conversion formula, the transformation formula is derived from the definition of mean and variance of negative binomial distribution in this study.

$$E[x] = \sum x \cdot nbin(x, a, b) = a \cdot \frac{1-b}{b}, Var[x] = a \cdot \frac{1-b}{b^2} \quad (21)$$

Then we can get:

$$a = \frac{E[N_{0,j}]^2}{Var[N_{0,j}] - E[N_{0,j}]}, b = \frac{E[N_{0,j}]}{Var[N_{0,j}]} \quad (22)$$

In this case, we only need to satisfy the requirement that the variance of repairs is greater than its mean value, which is obvious. According to Equations (22) and (20) for calculating EBO can be converted into:

$$EBO(s_{0,j}) = \sum_{x=s_{0,j}+1}^{\infty} (x - s_{0,j})nbin(x, a, b) = \sum_{x=s_{0,j}+1}^{\infty} (x - s_{0,j}) \binom{a + s_{0,j} - 1}{s_{0,j} - 1} b^{s_{0,j}}(1 - b)^a \quad (23)$$

When the system is a series structure, the system's availability is the product of all LRU's availability:

$$A(s_{0,j}) = \left(1 - \frac{EBO(s_{0,j})}{M_{0,j}}\right)^{M_{0,j}} \quad (24)$$

The marginal benefit factor $\delta$ is used to complete the optimization solution of spare parts inventory through the marginal analysis algorithm, and the solution logic is shown in Figure 3.

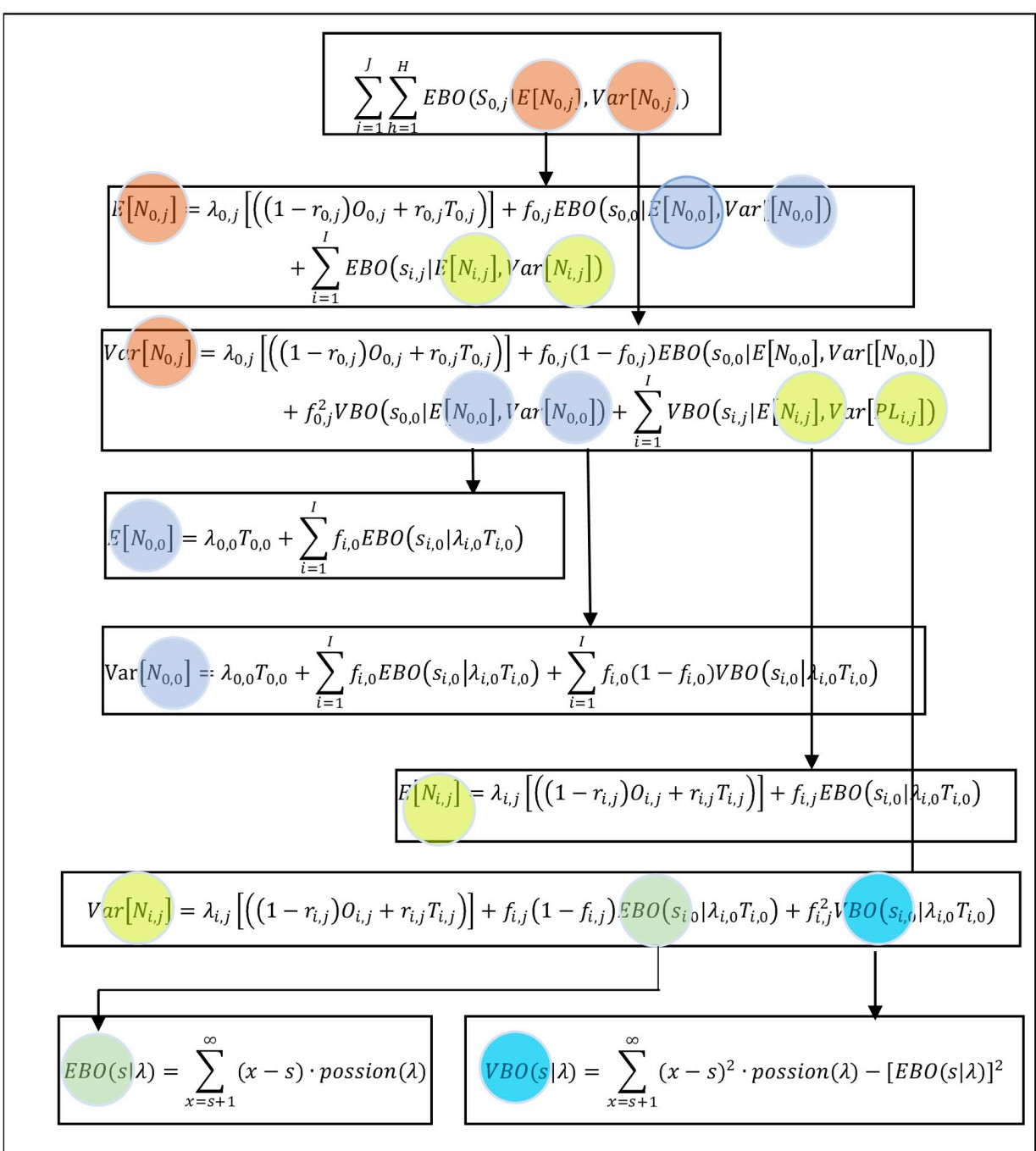

**Figure 3.** System EBO solution process.

### 3.3. Mathematical Model of Joint Optimization

Based on the above service logistics model, a two-echelon two-indenture nonlinear joint optimization model of fleet system is constructed by introducing multiple failure modes and discard decision.

$$
\begin{aligned}
\min \ & \sum \sum fc_i \cdot s_{i,j} + \sum \sum \sum vc_{i,j,d} \cdot X_{i,j,d} + \sum \sum rc_{r,j} \cdot Y_{r,j} \\
s.t. \quad & A\left(s_{i,j}\right) \geq A_{min}, i= 0,1,\cdots, j= 0,1,\cdots \\
& X_{0,j,repair} + X_{0,j,move}= \lambda_{0,j}, j= 1,2,\cdots \\
& X_{0,0,repair} + X_{0,0,discard}= \sum_{j=1}^{J} X_{0,j,move} \\
& X_{i,j,repair} + X_{i,j,move}= \lambda_{i,j} \\
& \sum_{j=1}^{J} X_{i,j,move}= \sum_{d \in D} X_{i,0,d} \\
& X_{0,j,repair}= \sum_{i=1}^{I} X_{i,j,repair}, i \in \Gamma_0 \\
& X_{i,j,repair}\leq Y_{r,j} \cdot \lambda_{i,j}, i \subset \Omega_r \\
& X_{i,j,d} \in N, Y_{r,j} \in \{0,1\}
\end{aligned}
\tag{25}
$$

The objective function consists of three parts: spare parts cost, maintenance decision cost, and maintenance resource cost. The constraints include two parts: availability constraints related to spare parts and LORA related to failure components.

If multiple failure modes are included in the LORA model, with the increase of component types, more detailed failure mode information is needed as the input of the model. The scale of model solution would be more and more difficult to the solute.

In the actual project, the serious failure mode usually requires more maintenance resources. In other words, the move ratio R is: $R = n/N$. Where $n$ is the number of moves and $N$ is the number of failed components.

The move ratio parameter R reflects the maintenance ability r of the Base locations, and its numerical relationship is:

$$
r = 1 - R = 1 - n/N = X_{i,j,move}/\lambda_{i,j}, j = 1,2,\cdots
\tag{26}
$$

Discard decision $X_{i,0,discard}$ means the system needs to order spare parts. Spare parts order lead time is $L_i$, the probability of components being discarded is $r_{i,0}$, the number of repairs at Depot is:

$$
E[N_{i,0}] = m_{i,0}[((1 - r_{i,0})T_{i,0} + r_{i,0}L_i)], i = 0,1,\cdots I
\tag{27}
$$

where $r_{i,0} = X_{i,0,d}/\lambda_{i,j}$.

To summarize, there would be $r$ of failure components repaired at Base location; $r_{i,0}$ of failed components discarded due to the high maintenance cost; and $1 - r - r_{i,0}$ of failed components repaired at Depot location.

Some authors focus on how to assess system availability with the constraint of resources, especially spare parts and equipment, by analytical method or numerical approach [31–33]. The practical motivation of this research is optimizing stock allocation targeting maximum availability with the constraint of preassigned maintenance policy. The essential motivation of this technique is on the hypothesis that availability can be dividing into two independent parts, maintenance availability and supply availability, respectively. So, most of the research express the effect by EBO or MDT (mean delay time). However, different maintenance capability factors indicate that maintenance decision-making changes dynamically. The approach of this airtcle tries to define the logistic support to ensure a specified availability. In this case, an availability-cost function is created in order to evaluate service logistics costs associated with a required service level.

The better the maintenance capability of the Base level, the better the system availability. System availability (A) is proportional to maintenance capability ($r$). Due to the maintenance resources allocation, the service logistics cost (C) is negatively correlated with $r$. In summary, A is negatively correlated with C. As two important indicators of the service logistics system, A and C can be connected with each other through $r$ of Base location.

### 3.4. Mathematical Model Solution

In reality, the joint solution of LORA and spare parts stocking is a sequential optimization. First, the costs with failure number and repaired costs can be optimized by LORA. Then, the spare parts stocking can be dealt with VARI-METRIC, based on the LORA's results. Finally, the location of spare parts in the repair network can be found to achieve a target availability of capital goods with the lowest possible spare parts holding costs [30].

The proposed iterative algorithm is shown in Figure 4. It is obvious to find that the sequential approach is two of the same building blocks.

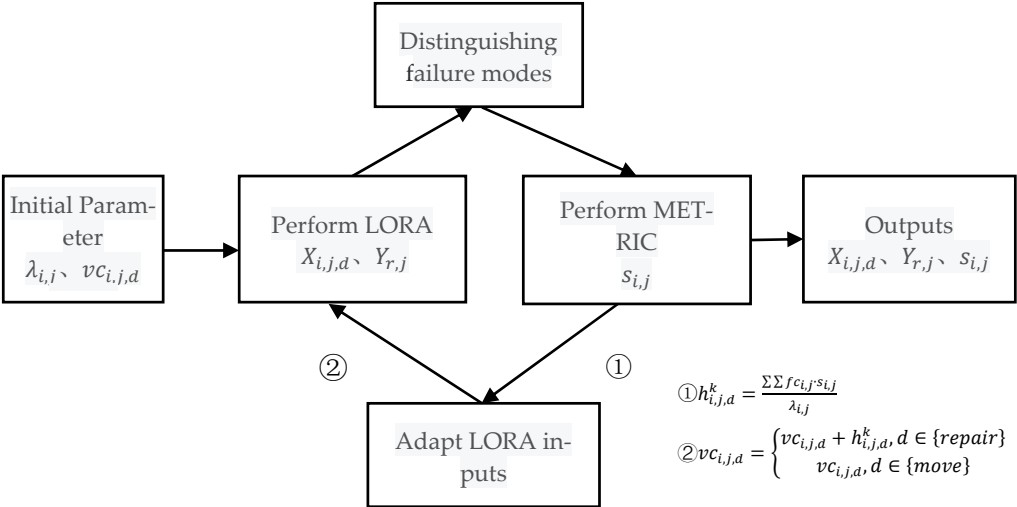

**Figure 4.** Joint iterative optimization method.

By solving the LORA model, the cost (maintenance resources cost $\sum \sum rc_{r,j} \cdot Y_{r,j}$ and maintenance cost $\sum \sum \sum vc_{i,j,d} \cdot X_{i,j,d}$) is minimized when the solution $\left( X_{i,j,d}, Y_{r,j} \right)$ satisfies the constraint conditions. According to the results, the maintenance ability $r$ of the component at the Base location is calculated. $r$ is taken as the input parameter of the spare parts inventory METRIC model, and the inventory solution $S_{i,j}$ is obtained.

The "intermediate" decision contains move decision, while the "final" decision includes repair and discard decision. Thus, only the costs of final decision should be focused on. We define $h_{i,j,d}^k$ as the decision cost of component $i$ added to LORA model in the $k$-th iteration:

$$h_{i,j,d}^k = \frac{\sum \sum fc_{i,j} \cdot s_{i,j}}{\lambda_{i,j}} \tag{28}$$

where $s_{i,j}$ is the result from iteration $i - 1$.

The cost coefficient of maintenance decision is:

$$vc_{i,j,d} = \begin{cases} vc_{i,j,d} + h_{i,j,d}^k, d \in \{repair\} \\ vc_{i,j,d}, d \in \{move\} \end{cases} \tag{29}$$

As mentioned above, the holding costs as the inputs of LORA should be changed in $i$-th iteration only if the corresponding repair/discard decision was selected in the former iteration. This is a way to gradually estimate the resultant holding costs for all

corresponding repair/discard decision. Then, the proposed algorithm will find optimal total costs: LORA costs and holding costs resulting from the spare parts stocking analysis.

The stop condition of the proposed algorithm is the solution of LORA and is identical to the previous iteration. If there are two different LORA solutions with the same costs, the solution shown in both iterations is selected, and the algorithm terminates.

## 4. Numerical Experiment

We take two key subsystems—LRU1 and LRU2—of the fleet system in reference [30] as examples. LRU1 consists of four SRUs; LRU2 consists of three SRUs. Thus, we can verify the effectiveness and correctness of the joint optimization model through considering 7 different failure modes. The two-echelon service logistics system consists of one Depot location and three Base locations. The number of subsystems in the three Base locations are 13, 30, and 20. The annual demand of spare parts for the two key LRUs at the three Base locations are 3, 7, 5, and 2, 4, 3. The target availability of the fleet system is 95%.

### 4.1. Analysis of the Influence of Maintenance Capability on Service Logistics Cost

Considering the change of component maintenance decision caused by multi failure modes, part of the failed components are left in the Base locations, and part of the failure parts are moved to the Depot locations to be repaired or discarded. The relationship between the maintenance capability at the Base locations and the total cost of the system under the constraint of 95% availability can be seen below in Figure 5.

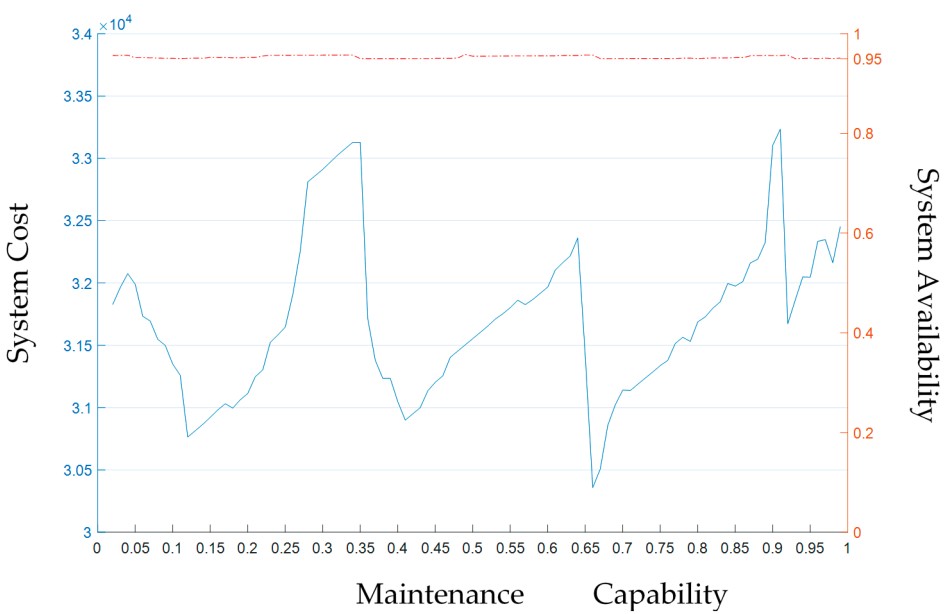

**Figure 5.** Joint iterative optimization method.

With the maintenance capacity gradually increasing from 0 to 1, the cost of the service logistics system presents a fluctuating trend. Under the constraint of availability, the optimization results of maintenance capability factor change dynamically. When the maintenance capacity factor is 0.65, the cost reaches the minimum value of 30,358,050. Lower maintenance capacity requires Depot locations to allocate more maintenance resources, and higher maintenance capacity would lead to an increase in maintenance activity costs. Therefore, we can confirm that the fleet system does exhibit an optimal maintenance capacity factor.

### 4.2. Spare Parts Inventory Analysis of Fleet System

Table 2 shows the inventory optimization results of the system when the maintenance capacity is 0.65 under the constraint of 95% system availability.

**Table 2.** Inventory optimization results.

| Components | Inventory Lever | | | |
|---|---|---|---|---|
| | **Depot** | **Base1** | **Base2** | **Base3** |
| $LRU_{1,0}$ | 4 | 9 | 17 | 15 |
| $SRU_{1,1}$ | 4 | 3 | 5 | 4 |
| $SRU_{1,2}$ | 2 | 1 | 2 | 2 |
| $SRU_{1,3}$ | 3 | 1 | 2 | 2 |
| $SRU_{1,4}$ | 4 | 2 | 5 | 3 |
| $LRU_{2,0}$ | 2 | 5 | 8 | 6 |
| $SRU_{2,1}$ | 2 | 2 | 3 | 3 |
| $SRU_{2,2}$ | 3 | 3 | 5 | 5 |
| $SRU_{2,3}$ | 3 | 3 | 5 | 5 |

By comparing the inventory results in reference [30], it is found that the optimization results reduce the inventory of LRU2 in our study with the inventory of LRU2's SRU increased slightly. Obviously, dynamic optimization of maintenance capacity could get more economical service logistics decisions.

In terms of maintenance decisions, they are more inclined to repair LRU by replacing SRU, which reduces the cost of purchasing the LRU2. At this point, the inventory of LRU1 is appropriately increased. In comparison, we saved 6.4% on spare parts inventory cost.

To illustrate the correctness of the model and algorithm of this study, we take the last iteration of the joint optimization model as an example. The inventory optimization process is shown in Figures 6 and 7.

Figure 6 shows the relationship between system EBO, availability, and system cost. System cost increasing means the inventory is increasing, the system availability is increasing, and the EBO is decreasing. According to the downward trend of the EBO curve, it can be judged that this curve conforms to the definition of the "convex" function; that is, any point on the curve is optimal. According to the relationship between availability and EBO, any point on the available curve line is also optimal.

Figure 7 shows the selection process of SRU, LRU1, and LRU2. The abscissa represents the number of iterations. If one iteration is completed, an additional inventory of parts is added. It can be seen in the first 40 iterations that the algorithm selects increasing the inventory of SRU as the optimal solutions.

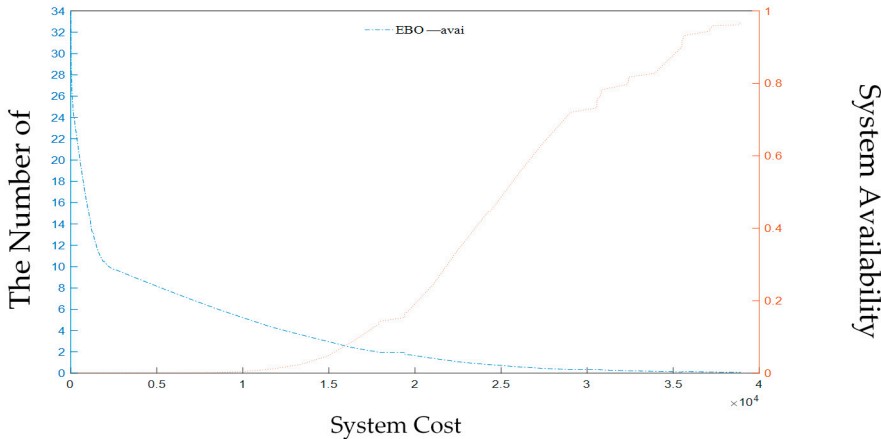

**Figure 6.** Evolution of EBO, system cost, and system availability.

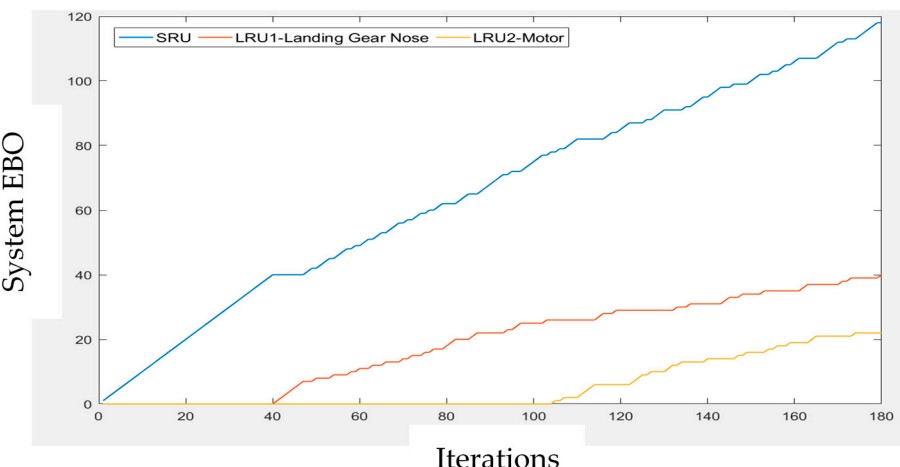

**Figure 7.** Component selection process.

According to the marginal benefit factor analysis, the factor value of SRU was greater than LRU1 and LRU2 in the first 40 iterations. Furthermore, since the price of SRU is much lower than LRU, the impact of price factor is greater than EBO reduction in the first 40 iterations.

When the iterations exceed 40, the algorithm selects increasing the inventory of LRU. This is due to the inventory of the SRU satisfying demand. At this time, $EBO_{SRU}(S_{i,j}|mT)$ is small enough, and the price of LRU1 is much higher than SRU. Therefore, $\delta_{SRU}$ is close to $\delta_{LRU1}$.

Similarly, since the price of LRU2 is much higher than LRU1 and SRU, when the iterations increase to 105, $\delta_{LRU2}$ is close to $\delta_{SRU}$, $\delta_{LRU1}$.

The same curve trends are also found in reference [30]. This proves the correctness of our model and algorithm. All cases that conform to the two-echelon two-indenture structure are applicable to this model and algorithm.

## 5. Conclusions

By considering the multiple failure modes, the new joint optimization model of repair level and spare parts inventory for a two-echelon two-indenture fleet system is built. By introducing maintenance capacity factor and discard decisions into the joint model, an improved iterative algorithm is designed to obtain the optimal solution of maintenance capacity and spare parts inventory. The effectiveness of the model and algorithm is verified by a typical example.

The contributions and innovations of this study are as follows: (1) The bridge of the dynamic maintenance capability factor between system availability and system cost is established effectively to solve the maintenance decision difficulties of increasing scales coming from the different characteristics of multiple failure modes considered in the LORA model. (2) Optimizing maintenance capability factor could dynamically get more economical joint optimization service logistics decisions. (3) The iterative process of the improved iterative algorithm shows that the availability of the fleet system can be effectively guaranteed by keeping the basic inventory of LRU unchanged and increasing the inventory of SRU. This study realizes the practical significance that the joint optimization model can guarantee the economy of the system.

As a further improvement, the risk of obsolescence of repairable systems can be introduced. Since the lifespan of the model is very wide, asset management related to life cycle cost is also factored into maintenance decisions and inventory management. For the additional multi-state components [34], common cause failures (CCFs) [35] and data uncertainty are complex problems regarding service logistics decisions.

**Author Contributions:** Conceptualization, R.W. and G.C.; methodology, J.W.; software, R.W.; validation, R.W., G.C. and Z.H.; formal analysis, R.W.; investigation, J.W.; resources, W.Z.; data curation, J.W.; writing—original draft preparation, R.W.; writing—review and editing, R.W.; visualization, R.W.; supervision, G.C.; project administration, G.C.; funding acquisition, W.Z. All authors have read and agreed to the published version of the manuscript.

**Funding:** This research received no external funding.

**Institutional Review Board Statement:** This study did not involve humans or animals.

**Informed Consent Statement:** Informed consent was obtained from all subjects involved in the study.

**Data Availability Statement:** Values of each variable at each site are presented in Tables 1 and 2 from literature [30]. The link of publicly archived datasets analyzed or generated during the study is https://pan.baidu.com/s/114wfw60oRjFdefPeFxHeDw (accessed on 6 August 2021). Extraction code: b3iv.

**Acknowledgments:** The authors gratefully acknowledge the support of The Key Projects of National Natural Science Foundation of China (No. 7153003), China National Defense Basic Science Research Program, and Application of innovation promotion in service base construction in Sichuan Province of China (No. 2017IM010700).

**Conflicts of Interest:** The authors declare no conflict of interest.

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
