# Peer review of "Joint Optimization Method of Spare Parts Stocks and Level of Repair Analysis Considering the Multiple Failure Modes"

_applsci, doi:10.3390/app11167254_

Round 1

Reviewer 1 Report

In this manuscript, the authors propose a joint optimization method to minimize the service logistics cost under the constraints of system availability and establish the bridge relationship between LORA and METRIC models. From the Comparison with the optimization of spare parts inventory and maintenance level independently, the joint optimization method could effectively reduce the service logistic system cost. It seems a very good idea. However, there are some suggestions for authors. The comments are as follows:

  1. Is there a specific relationship or photograph between system availability and system cost? This manuscript lists new joint optimization model, but it does not have a clearly conclusion about the impact of the dynamic maintenance capability factor and maintenance decisions according to the different characteristics of multiple failure modes

  1. The authors are supposed to provide more examples for their model. The authors described that optimizing maintenance capability factor dynamically could get more economical joint optimization service logistics decisions. If the example in the manuscript is enough to confirm their model under different failure modes?

Author Response

Q1: Is there a specific relationship or photograph between system availability and system cost? This manuscript lists new joint optimization model, but it does not have a clearly conclusion about the impact of the dynamic maintenance capability factor and maintenance decisions according to the different characteristics of multiple failure modes.

A:Thank you very much for your comments. The newly added literature [31][32][33] explains the relationship between availability, maintenance strategy and cost. It could improve the practical significance of the joint optimization model. See section 3.3 model description for details. And, the bridge of the dynamic maintenance capability factor between system availability and system cost is established effectively to solve the maintenance decision difficulties of increasing scales coming from the different characteristics of multiple failure modes considered in the LORA model. See section 5 for details.

Q2: The authors are supposed to provide more examples for their model. The authors described that optimizing maintenance capability factor dynamically could get more economical joint optimization service logistics decisions. If the example in the manuscript is enough to confirm their model under different failure modes?

A:Thank you very much for your comments. The structure of this system and the structure of service logistics system are both typical structures. The case of literature [30] conforms to the two-echelon two-indenture structure. Any other cases that conform to this structure are all suitable for our model. Such as the cases of literature [13] or literature [19]. In this manuscript, the analysis of the example verifies the correctness of the model and algorithm, and all cases that conform to the two-echelon two-indenture structure are applicable to our method and model. See section 4 for details.

Reviewer 2 Report

This article is at a good scientific level and contains all the necessary parts. The introduction to the research topic is well prepared and understandable and it is clear to see the motivation of this article. The optimization model is clearly described and each condition is justified. The article also contains experiments that are clearly and concisely justified. The conclusion is at a good level. As for the English language and style,  I do not feel qualified to judge, but still I was able to find some typos ( for example in 3.2. Mathematical model of spare parts stock: levles->levels).

Author Response

Q: This article is at a good scientific level and contains all the necessary parts. The introduction to the research topic is well prepared and understandable and it is clear to see the motivation of this article. The optimization model is clearly described and each condition is justified. The article also contains experiments that are clearly and concisely justified. The conclusion is at a good level. As for the English language and style, I do not feel qualified to judge, but still I was able to find some typos ( for example in 3.2. Mathematical model of spare parts stock: levles->levels).

A:Thank you very much for your comments. We have carefully checked and revised the grammatical text of the manuscript.

Reviewer 3 Report

The paper presents an interesting topic. It is well designed and appropriate for the readership of the Journal. However, it is necessary to bring some modifications before publishing it. They are discussed below:

It seems that the reference [is cancelled in accordance to the site:

  REV D CANCELLATION 1 - CANCELLATION NOTICE 1, REV D - Feb. 14, 1997 (https://www.document-center.com/standards/show/MIL-STD-1390/history/ )

The authors should update this reference.

The maintenance process and decision making in maintenance is not an isolated process. It is part of a larger concept known as asset management where maintenance is only a part of it. The authors should introduce a section upon asset management, the role and importance of maintenance in asset management and how the proposed approach integrated in the holistic asset management.

Author Response

Q1: It seems that the reference [is cancelled in accordance to the site: REV D CANCELLATION 1 - CANCELLATION NOTICE 1, REV D - Feb. 14, 1997 (https://www.document-center.com/standards/show/MIL-STD-1390/history/ ) The authors should update this reference.

A:Thank you very much for your comments. We have learned that MIl-STD-1390D has been replaced by GEIA-STD-0007. In LORA, GEIA-STD-0007 basically uses the relational tables in MIL-STD-1390D, and the definition of data units is basically the same. We have updated the literature [5].

Q2: The maintenance process and decision making in maintenance is not an isolated process. It is part of a larger concept known as asset management where maintenance is only a part of it. The authors should introduce a section upon asset management, the role and importance of maintenance in asset management and how the proposed approach integrated in the holistic asset management.

A:Thank you very much for your comments. Maintenance decision is an important part of the asset management of large-scale system, which belongs to the research category of the life cycle cost (LCC). Joint optimization method of spare parts stocks and level of repair analysis considering LCC plays an important role in our future research. See section 5 for details.